# Passive mode-locking and terahertz frequency comb generation in resonant-tunneling-diode oscillator

Tomoki Hiraoka [1✉], Yuta Inose[1✉], Takashi Arikawa [1,2], Hiroshi Ito[3] & Koichiro Tanaka [1✉]

Optical frequency combs in the terahertz frequency range are long-awaited frequency standards for spectroscopy of molecules and high-speed wireless communications. However, a terahertz frequency comb based on a low-cost, energy-efficient, and room-temperature-operating device remains unavailable especially in the frequency range of 0.1 to 3 THz. In this paper, we show that the resonant-tunneling-diode (RTD) oscillator can be passively mode-locked by optical feedback and generate a terahertz frequency comb. The standard deviation of the spacing between the comb lines, i.e., the repetition frequency, is reduced to less than 420 mHz by applying external bias modulation. A simulation model successfully reproduces the mode-locking behavior by including the nonlinear capacitance of RTD and multiple optical feedback. Since the mode-locked RTD oscillator is a simple semiconductor device that operates at room temperature and covers the frequency range of 0.1 to 2 THz (potentially up to 3 THz), it can be used as a frequency standard for future terahertz sensing and wireless communications.

[1] Department of Physics, Graduate School of Science, Kyoto University, Kyoto, Sakyo-ku 606-8502, Japan. [2] PRESTO, Japan Science and Technology Agency (JST), 4-1-8 Honcho, Kawaguchi 332-0012, Japan. [3] Center for Natural Sciences, Kitasato University, Sagamihara, Minami-ku 252-0373, Japan. ✉email: t.hiraoka1023@gmail.com; inose.yuta.t14@kyoto-u.jp; kochan@scphys.kyoto-u.ac.jp

The optical frequency comb is a crucial light source for metrology and spectroscopy. Its spectrum consists of equidistant optical modes[1]. The frequency of each mode is represented as follows:

$$f_n = f_{CEO} + nf_{rep}. \qquad (1)$$

Here, $f_{rep}$, $f_{CEO}$, and $n$ are the repetition frequency, carrier-envelope-offset frequency, and modal index, respectively. The optical modes are coherent and have a stable phase relationship with each other. The frequency-comb source is long-awaited as the frequency standard for spectroscopy of gaseous molecules[2] and high-speed wireless communications in the terahertz frequency range[3]. However, such light sources typically depend on energy-consuming and expensive femtosecond lasers[4]. The development of an energy-efficient and low-priced terahertz frequency-comb source based on a semiconductor device is still being pursued.

A promising candidate for a semiconductor-based terahertz frequency-comb source is the quantum cascade laser (QCL)[5], which is a compact device emitting watt-class terahertz waves[6,7]. A frequency comb using a terahertz QCL was recently demonstrated[8–13]. Moreover, differential frequency generation in mid-infrared QCL comb has been used to make a comb from 1.8 to 3.3 THz at room temperature[14,15]. However, it is difficult for a QCL to generate a terahertz comb below 1.8 THz. There are also devices based on Si CMOS technologies. For instance, a frequency-comb source based on a multiplier was demonstrated for spectroscopy in the range from 220 to 330 GHz[16]. Moreover, a bipolar CMOS device was used to generate a frequency comb from 0.03 to 1.1 THz[17]. However, it is difficult for CMOS devices to generate terahertz waves of higher frequency.

This study reports a terahertz comb source, a passive mode-locked resonant-tunneling-diode (RTD) oscillator. The RTD oscillator is an electrical device with a fundamental oscillation frequency in the terahertz frequency range at room temperature[18]. Oscillation from the sub-terahertz to 1.98 THz range has been achieved[19–23], and oscillation up to 2.77 THz is expected[24]. A single oscillator can be fabricated on a millimeter-sized chip[25]. The emission power reached 0.4 mW for a single oscillator at 530–590 GHz[26] and 0.73 mW for a large-scale array at 1 THz[27]. The DC-to-RF conversion efficiency in the terahertz region is about 1%[26]. However, there has been no report on mode-locking and frequency-comb generation in an RTD oscillator. In this paper, we show that the RTD oscillator can be passively mode-locked by simply controlling the optical feedback and that a terahertz frequency comb can be generated. We also demonstrate that the repetition frequency can be stabilized by external modulation. We present a simulation model which reproduces the mode-locking and predicts the future improvement in comb performance.

## Results

**Measurement of the emission spectrum**. Figure 1a is a schematic diagram of the experimental setup. We measured the emission spectrum of an RTD oscillator under optical feedback with variable amplitude and delay. The distance between the oscillator and the mirror $z_M$ was about 500 mm. We performed a heterodyne measurement with the local oscillator (LO) signal, which had a center frequency of 303.5 GHz and a linewidth of less than 240 mHz at FWHM (see the Experimental setup section in the Methods). Figure 1b shows a typical emission power spectrum of a continuous-wave (CW) oscillatory state observed without optical feedback from the mirror. It is a single-frequency spectrum with minor sidebands with much lower power spectral densities (PSD) compared with the main peak. The bottom axis

shows the heterodyne frequency, and the top axis shows the corresponding terahertz frequency.

We found that a frequency comb is generated when optical feedback is injected into the RTD oscillator in a certain phase. The red trace in Fig. 1c shows a typical frequency-comb spectrum. Including the small peaks that are not numbered in Fig. 1c, there are optical modes with a mode spacing of 273.3 MHz. The mode spacing was approximately proportional to the inverse of $z_M$, showing that optical feedback from the mirror causes the optical modes. We note that the mode spacing is smaller than the free-spectral range of a Fabry-Perot cavity with the cavity length of $z_M$, i.e., $c/2z_M$ (~300 MHz for $z_M \sim 500$ mm), where $c$ is the speed of light. This is because our setup is in a weak feedback regime (due to the low antenna coupling efficiency), which is different from the Fabry-Perot cavity corresponding to the strong-feedback limit (see Supplementary Note 2). It is also confirmed by the numerical simulation shown later. The spectrum of Fig. 1c is a harmonic frequency comb[28,29], where the spacing of the strongly oscillating modes, i.e., the repetition frequency, is an integer multiple of fundamental mode spacing. In the present experiment, the repetition frequency of 4 mode spacing was the most stable. The RF frequencies of the numbered peaks are described with the following equation:

$$f_n^{RF} = f_0^{RF} + nf_{rep}. \qquad (2)$$

Here, $f_n^{RF}$ is the RF frequency of the mode with index $n$, $f_0^{RF}$ is the offset RF frequency, and $f_{rep}$ is the repetition frequency of the harmonic comb. We fitted the relationship between the frequencies of the comb lines $f_n^{RF}$ ($n = 0$–9) and $n$ with Eq. (2) and obtained the parameters with the average values and standard deviation as follows: $f_0^{RF} = 618.97 \pm 0.45$ MHz and $f_{rep} = 1093.13 \pm 0.11$ MHz (see the Spectrum characterization in the Methods).

The peaks shown in the black trace of Fig. 1c are homodyne signals that appeared even when we blocked the LO signal. Figure 1d shows the homodyne signal measured under the same conditions as those of Fig. 1c. There are three peaks, and their frequencies match integer multiples of $f_{rep}$ within the margin of error. Hence, the homodyne peaks are the inter-mode beat note of the harmonic comb. Figure 1e and f shows the magnified spectrum of the comb line indexed as $n = 3$ and the homodyne peak at 1.0931 GHz. The linewidth of the comb line is 1.9 MHz. The homodyne peak has a smaller linewidth of 310 kHz. Its small linewidth corresponds to a small error in $f_{rep}$ and implies that the optical modes are phase-locked to each other.

**Relative modal phases**. To clarify that the modal phases obey a stable relationship, we measured the single-shot temporal waveform of the heterodyne signal shown in Fig. 1c. A sequential waveform was measured over 65.6 μs, as shown in Supplementary Note 3. The dots in Fig. 2a show a typical part of the measured waveform, and the trace shows a fitting curve obtained in the analysis below. The heterodyne waveform has an average period of ~200 ps corresponding to the center RF frequency of 5 GHz in the comb spectrum. We can also see that it has a frequency modulation with a period of ~1 ns, reflecting the $f_{rep}$ of about 1 GHz. This frequency-modulated property is further clarified in the analysis below.

We performed a fitting analysis of the heterodyne waveform to clarify the phase relationship between the modes. We utilized a fitting function representing the heterodyne beat of the frequency

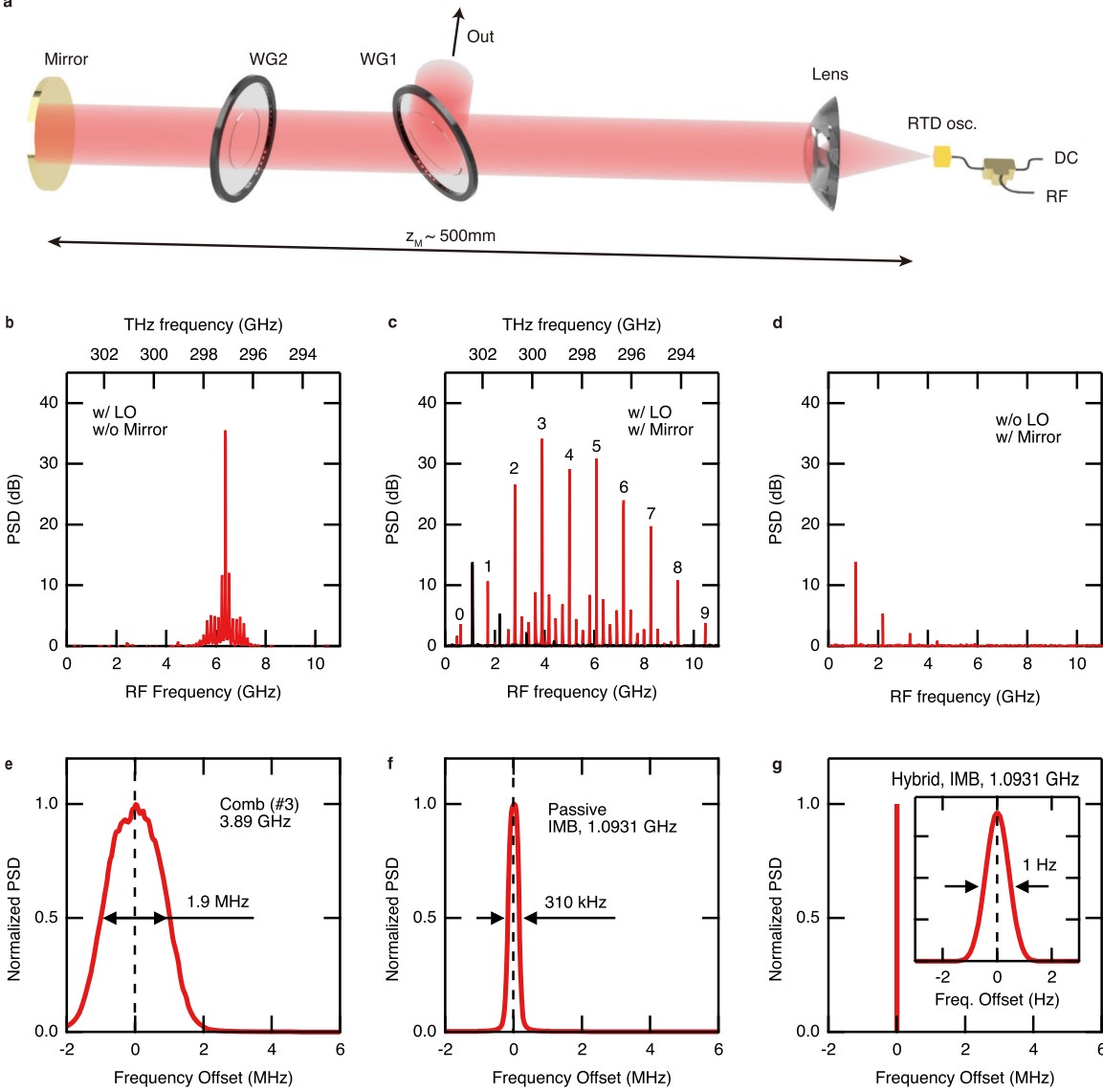

**Fig. 1 THz frequency-comb generation in RTD oscillator. a** Schematic diagram of the experimental setup. The RTD oscillator is biased with a DC bias voltage and generates a terahertz wave. We applied an external modulation only when we demonstrated hybrid mode-locking. The terahertz emission is split into two beams by the wire-grid polarizer WG1 with a power ratio of 1:1. The beam transmitted by WG1 is reflected at the mirror and fed back to the RTD oscillator. The distance between the mirror and the oscillator, $z_M$, is about 500 mm. It is tunable with a motorized stage on which the mirror is mounted. The amplitude of the return light is controlled by rotating another wire-grid polarizer, WG2. WG2 is tilted to the beam in order to prevent a direct reflection to the oscillator. The beam reflected at WG1 enters the measurement part. **b** Emission spectrum of CW oscillation state observed when the return light was blocked (w/o Mirror) and measured with the local oscillator signal (w/ LO). The left axis shows the power spectral density (PSD) relative to the noise level. The bottom axis shows the heterodyne frequency, and the top axis shows the corresponding terahertz frequency. **c** Frequency-comb spectrum measured observed under optical feedback (w/ Mirror) and measured with the LO signal. The peaks shown by the black trace were observed even without the LO signal. The numbers at the peaks are the mode indices of the frequency comb. **d** Emission spectrum of the passive mode-locked state measured without the LO signal (w/o LO). Three peaks are inter-mode beat notes. **e** Magnified view of a comb line indexed as $n = 3$. The vertical axis is PSD normalized with the peak height. **f** Magnified spectrum of the inter-mode beat note (IMB) at 1.0931 GHz. **g** Magnified spectrum of the inter-mode beat note at 1.0931 GHz when the bias modulation was applied (hybrid mode-locked state). These spectra were accumulated over 1 second. The bias voltage was 471 mV.

comb:

$$f(t) = \sum_{n=2}^{6} A_n \sin\left[2\pi\left(f_0^{\mathrm{RF}} + n f_{\mathrm{rep}}\right)(t - t_0) + \varphi_n\right]. \quad (3)$$

Here, $n$ is the modal index shown in Fig. 1c. We considered only the five modes of $n$ from 2 to 6, which have significant amplitudes. $A_n$ denotes the amplitudes of the modes, which are fixed parameters derived from the spectrum. $t_0$ is the time origin. $\varphi_n$ denote the initial phases at $t = t_0$. $f_0^{\mathrm{RF}}$, $f_{\mathrm{rep}}$, $t_0$, and $\varphi_n$ are the

fitting parameters. We neglected the phase fluctuation of the LO signal because its linewidth was less than 240 mHz (coherence time of over 4 seconds, which is much longer than the measured waveform). Note that, in the time scale defined by linewidths of the comb lines (1/1.9 MHz = 520 ns), noise causes a random phase shift in each mode. Hence, we cannot fit the waveform longer than this time scale with Eq. (3), in which each frequency component is described as a single sinusoidal wave with a well-defined phase. Thus, we divided the long span of 65.6 μs into

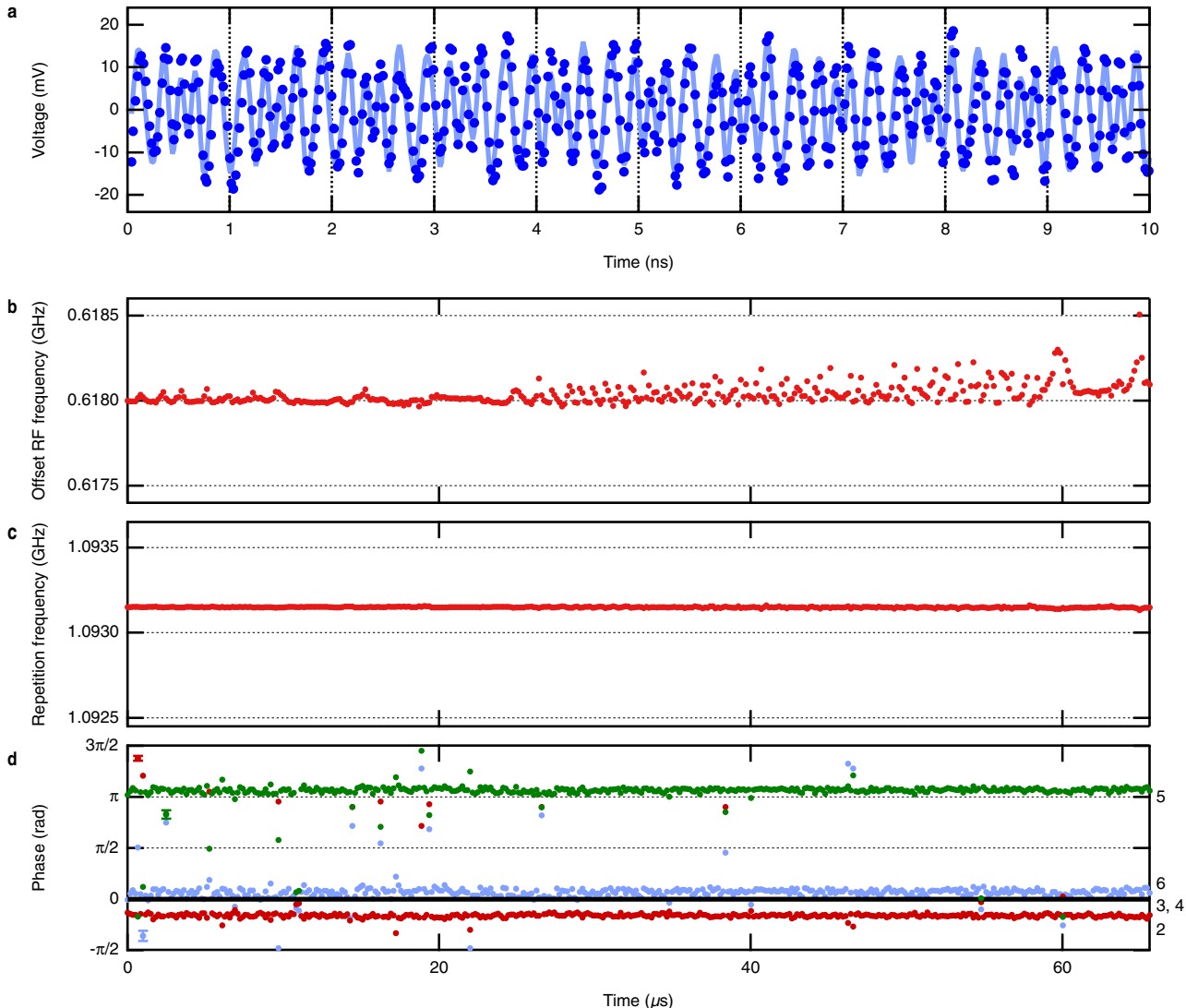

**Fig. 2 Fixed relationship between modal phases. a** Measured heterodyne temporal waveform of passive mode-locked state (dots) and fitting curve (trace) plotted over 10 ns. The temporal resolution of the measurement was 20 ps. Long-term stability of **b** offset frequency, **c** repetition frequency, and **d** relative phase offsets over 65.6 μs. The numbers beside the right axis show the mode indices corresponding to the markers. The error bars show the estimated standard deviation of the fitting parameter. In panels **b** and **c**, the error bars are smaller than the marker size. In panel **d**, the error bars are shown in one data point in the first few μs for each marker as a typical value.

short spans of 164 ns and fitted the waveform in each short span. These short spans had to be enough shorter than 520 ns, an averaged time scale where the noise is induced, to obtain good fitting convergence in as many short spans as possible. A typical fitting curve is shown as the trace in Fig. 2a; it fits the data points. It is not a short and intense pulse, as is often the case for a mode-locked pulse. We note that we carefully defined the time origin $t_0$ in each short span to represent the relationship between the initial phases $\varphi_n$ uniquely. We defined the time origin $t_0$ in each span as the time at which the condition $\varphi_3 = \varphi_4$ is satisfied. The details of the fitting are shown in Supplementary Note 4.

Figure 2b and c shows $f_0^{RF}$ and $f_{rep}$ for each fitting span. The average values and standard deviations considering the fitting error are as follows: $f_0 = 618.039 \pm 0.061$ MHz and $f_{rep} = 1093.1500 \pm 0.0032$ MHz. The average values are consistent with those derived from the spectrum. The standard deviations are smaller than the linewidths in Fig. 1e and f. It indicates that there is a long-term deviation not observed in this span.

We found that the relative initial phases $\triangle\varphi_n \equiv \varphi_n - \varphi_4$ are kept constant for 65.6 μs as shown in Fig. 2d, even though the initial phases $\varphi_n$ changes randomly in the time scale of 520 ns. This is evidence of the mode-locking, which stabilizes the phase relation between the modes even under the noise. Here, the noise causes only a timing jitter of the mode-locked waveform and a uniform phase shift over the modes. If there were no mode-locking effect, the relative initial phases would be randomized in the time scale of 520 ns. The average values and standard deviations of the relative initial phases are as follows: $\triangle\varphi_2 = -0.40 \pm 0.61$, $\triangle\varphi_5 = 3.29 \pm 0.48$, $\triangle\varphi_6 = 0.25 \pm 0.49$ rad. Their relation can be expressed approximately as

$$\left(\triangle\varphi_2, \triangle\varphi_3, \triangle\varphi_4, \triangle\varphi_5, \triangle\varphi_6\right) \cong (0, 0, 0, \pi, 0). \qquad (4)$$

We note that the relationship of Eq. (4) is different from that of the typical mode-locked lasers based on saturable absorbers. All the modes have the same phase in such lasers, and they show amplitude-modulated waveforms. The relationship of Eq. (4) rather means a frequency-modulated waveform, as described in

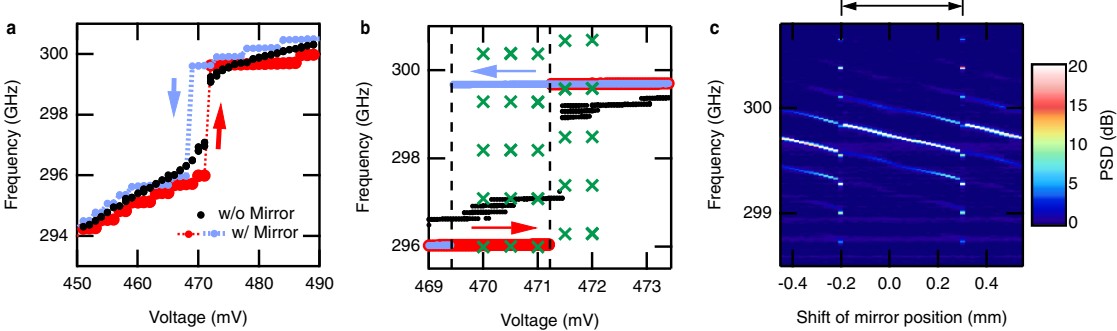

**Fig. 3 Conditions of passive mode-locking. a** Frequency-voltage curve measured without mirror (black dots), with mirror and up-swept voltage (red dots), and with mirror and down-swept voltage (blue dots). Significant hysteresis on the sweep direction was not observed in the case of no mirror. **b** Frequency-voltage curve measured around the frequency jump point (markers are the same as in **a**). The green crosses show the frequencies of the comb peaks observed when the mirror was swept at each bias voltage. **c** THz spectrum observed when the mirror was swept with a bias voltage of 471 mV. The horizontal axis shows the shift of the mirror position $\triangle z_M$. The sweep direction was the one in which $\triangle z_M$ decreases. These figures were measured with the maximum feedback amplitude in our setup. The frequency resolution was 11 MHz. The data points in panels **a** and **b** were extracted from the series of spectra obtained during the voltage sweep and are the frequency points that had a PSD larger than the noise level by 20 dB.

detail in Supplementary Note 5. Such a frequency-modulated waveform is also observed in the frequency comb generated in the QCL[30] and the mode-locked dark pulse generated in the microresonator[31].

**Conditions for passive mode-locking**. We found that passive mode-locking occurred only around a particular point in the frequency-voltage curve, which we call the "frequency jump." Figure 3a shows frequency-voltage curves measured with and without optical feedback from the mirror. When there is no optical feedback, the curve shows a frequency jump of about 2 GHz around 471 mV. The frequency changes continuously at the other bias points. When feedback is present, many small steps appear in the frequency-voltage curves. The oscillation frequency shows a hysteretic behavior in the sweeping direction. A large hysteresis loop in the frequency-voltage curve formed at the frequency jump point of 471 mV. These behaviors can be qualitatively explained with the oscillation condition for a simplified circuit model with optical feedback[32], which is given in Supplementary Note 2. Furthermore, these frequency-voltage curves were reproduced in a simulation, as shown in the next section. When the bias voltage is set near the frequency jump point and the position of the mirror is swept, the passive mode-locking state appears. Figure 3b shows the detailed frequency-voltage curve measured near the frequency jump and the peak frequencies of the comb (green crosses). We swept the mirror at bias voltages from 467 to 475 mV in 0.5 mV steps and obtained the comb spectra only in the range of 470–472 mV, which is the vicinity of the frequency jump.

The comb spectra appeared at a particular mirror position. Figure 3c shows the heterodyne spectrum measured by sweeping the mirror in steps of 0.02 mm at a fixed voltage of 471 mV. The comb spectra were observed periodically at the mirror position, as shown by the vertical lines on the top of Fig. 3c. The period was 0.500 mm. The round-trip length of 1.000 mm is equal to the wavelength of the terahertz wave of 300 GHz. This shows that passive mode-locking takes place at a certain phase of the optical feedback.

In the present experiment, the feedback amplitude was close to the lower limit of the passive mode-locking. When the feedback amplitude was reduced to less than 93% of the maximum amplitude, the passive mode-locked state disappeared. The details of the feedback amplitude dependence are described in Supplementary Note 6.

**Hybrid mode-locking**. We succeeded in stabilizing the repetition frequency of the frequency comb by using the hybrid mode-locking technique[33–35]. Here, we applied a bias-voltage modulation on the passively mode-locked RTD oscillator. The modulation frequency was set to 1.0932 GHz (with a linewidth of less than 1 Hz), the same as the repetition frequency of the passive mode-locked state, i.e., the harmonic-comb spacing and first inter-mode beat-note frequency. The output power of the modulator was only −40 dBm, while the emission power from the RTD oscillator was −20 dBm.

The frequency-comb spectrum in the hybrid mode-locked state is almost the same as the one obtained in the passively mode-locked state (see Supplementary Fig. 7-1), except for the significant reduction in the linewidth of the inter-mode beat note (Fig. 1g). By applying the modulation, the linewidth of the inter-mode beat note decreased to less than 1 Hz, which corresponds to the standard deviation of 420 mHz in the repetition frequency. We note that the linewidth of the comb lines did not change a lot from that of the passive mode-locked state (Supplementary Fig. 7-1), in contrast to that of the inter-mode beat note. This means that the hybrid mode-locking stabilizes the repetition frequency but does not stabilize the carrier-envelope-offset frequency. The details of the hybrid mode-locking and conditions for achieving hybrid mode-locking are described in Supplementary Note 7.

**A circuit model for the passive mode-locking**. Here, we present a circuit simulation model that reproduces the frequency-voltage curve, the frequency comb in the vicinity of the frequency jump, and the frequency-modulated waveform. The model simulates an LCR parallel circuit with an RTD. It includes not only the non-linear conductance but also the nonlinear capacitance of the RTD[36–38]. The optical feedback is included as feedback current $I_{FB} = \sqrt{\eta}I_{load}(t - t_d)$. Here, $I_{load}(t)$ is the current at the load in the circuit, and $t_d$ is the time delay. $\eta$ is a reflectivity including the coupling efficiency. Noise in the circuit is included. The circuit diagram and parameters are given in Supplementary Note 8.

Figure 4a shows a simulated frequency-voltage curve that reproduces the experimentally measured curve in Fig. 3a. To reproduce the frequency jump, we found that two additional optical-feedback terms with parameters $(t_d, \eta)$ of $(19.7\ ps, 10^{-2.0})$ and $(178\ ps, 10^{-3.0})$ were necessary [see Supplementary Note 8 (3-1)]. They correspond to reflection surfaces separated from the oscillator by 2.95 mm and 26.7 mm, presumably due to the device

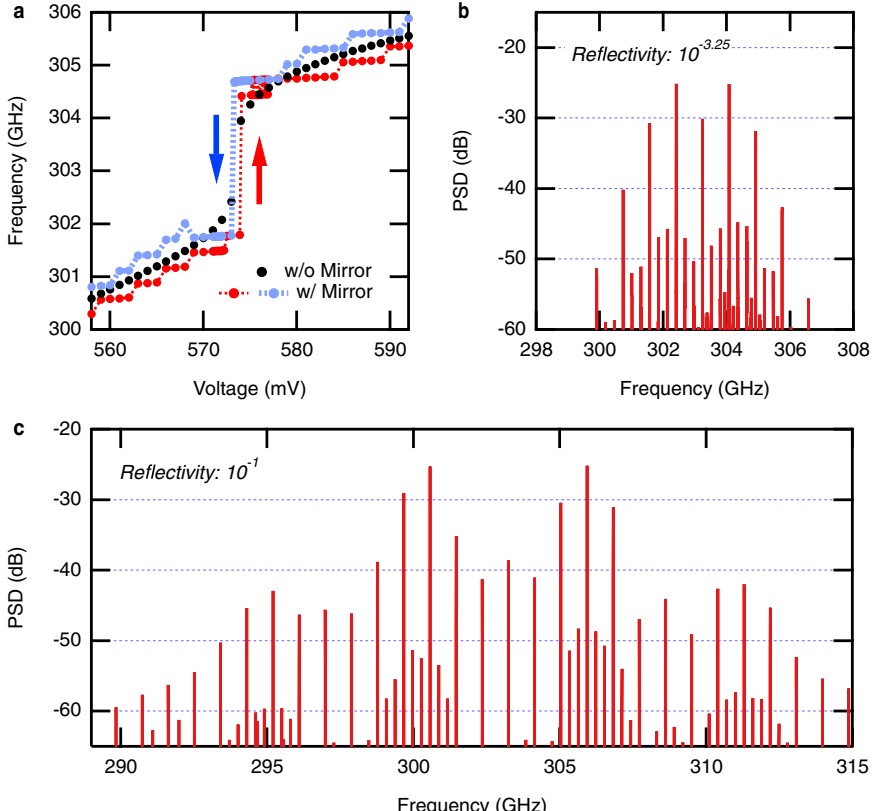

**Fig. 4 Circuit simulation. a** Wide-range frequency-voltage characteristics. In the simulation, the temporal waveform was calculated by sweeping the bias voltage. At each data point, the voltage sweep was stopped, and the temporal waveform was simulated for 0.62 μs. The spectrum was obtained by Fourier transforming the temporal waveform of the last 0.1 μs. The shot noise described in Supplementary Note 8 (1) was included. **b** Harmonic frequency-comb spectrum simulated for reflectivity $\eta_{Mirror} = 10^{-3.25}$, which corresponds to the experimental condition. **c** Harmonic frequency-comb spectrum simulated for $\eta_{Mirror} = 10^{-1}$. Since the circuit has a nonradiative loss, it corresponds to the case where all the emitted power is fed back from the mirror. The results in panels **b** and **c** obtained under the following conditions: the bias voltage was 573.5 mV. The temporal waveform was simulated over 11.0 μs, and the spectrum was calculated using the last 1.0 μs. Noise had a standard deviation 10 times smaller than the shot noise.

itself and the experimental setup. The time delay of the feedback from the mirror was set to 3340 ps, which corresponds to the distance of 500 mm, and the reflectivity was determined as $10^{-3.25}$ based on the mode spacing [see Supplementary Note 8 (3-2)]. Around the frequency jump, we found a state which produces a harmonic frequency-comb spectrum, as shown in Fig. 4b. Here, we note that the fundamental mode spacing in the simulation result is 278 MHz, which is smaller than the inverse of the time delay (1/3340 ps = 300 MHz.) This is because the small feedback amplitude results in the mode spacing smaller than the free-spectral range of Fabry-Perot cavity of the strong-feedback limit. The harmonic-comb spectrum was preserved under a noise level of one-tenth of the shot noise but was not preserved under the shot noise level. We could not verify whether the mode-locked state can be made more stable by tuning the parameters, or if we need another stabilizing effect. In Supplementary Note 9-1, we show that the temporal waveform is not an intensity-modulated waveform, but rather a frequency-modulated one. We expect that this oscillatory state corresponds to the passive mode-locked state in the experiment.

To investigate the mechanism of passive mode-locking, we performed a simulation experiment removing the nonlinear effects one by one from the conditions of Fig. 4b. When we removed the feedback term with a time delay of 19.7 ps and 178 ps, we obtained neither a frequency jump nor a comb spectrum. When we replaced the nonlinear capacitance with a constant capacitance of 8 fF, we obtained a frequency jump around 303 GHz, but no comb spectrum. On the other hand, we

obtained a comb spectrum when we removed the noise. Hence, feedback with a short delay time and a nonlinear capacitance are necessary for passive mode-locking, whereas noise is not necessary. As far as we know, the mode-locking caused by such effects is different from the conventional mode-locking mechanisms. It is a subject for future work to determine how these effects cause mode-locking.

## Discussion

Although some studies on the optical-feedback effect in RTD oscillators have been done, the mode-locking behavior has not been reported. There are studies showing that optical feedback affects the oscillation frequency and emission power, but their report was limited to single-mode oscillation[32,39]. There is another report implying self-pulsation due to optical feedback[40]. However, the conditions to obtain the self-pulsation and the mode-locking state have not been clarified. Pulsed emission can be obtained from RTD relaxation oscillators[41]. Although its spectrum consists of several phase-locked modes, it is simply due to the harmonic generation.

Finally, we discuss the improvement in comb performance of the mode-locked RTD oscillator. Through hybrid mode-locking, the repetition frequency can be tuned with an external signal. Therefore, if we can stabilize the carrier-envelope offset frequency, we can obtain a fully stabilized comb spectrum. To stabilize the offset frequency, a resonant-tunneling-diode oscillator combined with a varactor diode[42] would be effective. In this

oscillator, a phase-locked-loop (PLL) control through the varactor diode can be used to decrease the linewidth to less than 1 Hz in a CW oscillation state. Stabilization of one of the comb lines through PLL control would stabilize the offset frequency. Fixing one of the comb lines to a molecular absorption line will also result in narrow frequency-comb lines with known absolute frequencies.

The simulation model shows that we can broaden the spectral bandwidth of the frequency comb in a different feedback condition. Figure 4c shows a simulated harmonic frequency-comb spectrum with a larger feedback amplitude. In this case, the comb spectrum is broader than in Fig. 4b. We note that the temporal waveform is frequency-modulated and not amplitude-modulated in this case, too (see Supplementary Note 9-2.) The simulation shows that various broadband comb spectra can be generated depending on the feedback conditions (see Supplementary Note 10). It also showed the possibility to make a compact feedback configuration; optical feedback from surfaces with distances of 2.95 and 26.7 mm can cause the mode-locking even without the feedback from the mirror (see Supplementary Fig. 10b). Although we showed the possible improvement of the performance by the feedback parameters, investigation of the vast parameter space of the feedback conditions both in experiment and simulation is an important future task to understand how the spectral shape and bandwidth of the frequency comb are determined. It will help us to optimize the feedback conditions and circuit parameters to obtain the desired spectra in a compact setup.

In conclusion, we clarified that a terahertz frequency comb can be obtained from a passive mode-locked resonant-tunneling-diode oscillator. Mode-locking is achieved with controlled optical feedback. We succeeded in stabilizing the repetition frequency with an additional bias-voltage modulation. The mode-locked waveform was not a short and intense pulse but rather a frequency-modulated waveform. By including the nonlinear capacitance of RTD and multiple optical feedback, a simulation model reproduced several behaviors of mode-locking. It suggested the possibility of broadband comb generation and compact feedback configuration. A better understanding of the mechanism and engineering will lead to high-performance terahertz frequency-comb generation from an RTD oscillator. Since the mode-locked RTD oscillator is an energy-efficient and room-temperature operating semiconductor device, we believe it is suitable as a frequency standard for terahertz sensing and wireless communications.

## Methods

**Experimental setup**. A detailed schematic figure of the experimental setup is shown in Supplementary Note 1. The evaluated RTD oscillator is a prototype oscillator with a plastic leaded chip carrier package $4 \times 4 \times 2.44$ mm in size, made by Rohm Co., Ltd[25]. It was connected to a source meter and a signal generator via a bias-Tee. The RTD oscillator was biased with a DC voltage. When we wanted to show the effect of the bias modulation, we used a signal generator (RF002, RFnetworks Corporation). The signal generator was stabilized using the 10 MHz frequency reference from the atomic clocks in global positioning satellites (GPS). The current-voltage curve of the oscillator is shown in Supplementary Note 8 (1). The emission power was typically about 10 μW. We used a Tsurupica lens with a focal length of 70 mm (Tsurupica-R-CX-2.0-70-SPS, Pax. Co., Ltd.) as the terahertz lens in front of the RTD oscillator. The distance between the RTD oscillator and the mirror, $z_M$, was 500 mm, which is the shortest possible distance in our setup. When we increased $z_M$, the decrease of the comb bandwidth was observed as well as that of the mode spacing. We believe it is because the feedback amplitude decreased in the long distance due to the imperfect THz-wave alignment.

The measurement part is basically the same as that used in our previous study[43]. The local oscillator (LO) signal was a frequency-stabilized CW terahertz wave. We utilized a LO signal with a linewidth less than 240 mHz to evaluate the linewidth of the heterodyne spectrum and measure the temporal heterodyne waveform. The power of the LO signal was about 10 μW. The mixed terahertz wave was detected by a Fermi-level managed barrier diode (FMBD) with an amplifier bandwidth of 10 GHz[44]. The RF spectrum of the detected signal was measured with a spectrum analyzer (MXA 9020B, Keysight Technologies Inc). It had a bandwidth of 23 GHz and maximum resolution bandwidth of 1 Hz. The spectrum analyzer

was referenced to the 10 MHz frequency reference from GPS atomic clocks. The temporal waveform of the RF signal was also measured with an oscilloscope (MSO68B 10 GHz, Tektronix Inc). It had a sampling rate of 50 GS/s and a bandwidth of 10 GHz.

We should note that there would be some inaccuracy in the measured amplitude. The sensitivity of the measurement system might have some frequency dependence because of standing waves forming[45] between the oscillator and the detector. In addition, the mixed terahertz wave was so strong that saturation of the integrated amplifier in the FMBD module[44] might have taken place. Hence, it is difficult to compare the intensity of the frequency-comb spectrum and the inter-mode beat note. It is also difficult to discuss the depth of the amplitude modulation in the temporal waveform of the passive mode-locked state.

**Spectrum characterization**. In the evaluation of the comb lines, we derived the frequencies of the comb lines $f_n^{RF}$ ($n = 0$–$9$) as the center frequencies obtained by fitting the peaks with a Gaussian function. We fitted $f_n^{RF}$ with Eq. (2), taking the linewidths of the peaks as the standard deviation of $f_n^{RF}$.

Similarly, in the evaluation of the inter-mode beat notes, the frequencies of the three peaks, $f_{IMB,m}$, were derived from a Gaussian fitting. We fitted $f_{IMB,m}$ with

$$f_{IMB,m} = mf_{rep} \qquad (6)$$

where $m = 1$, 2, and 3, taking the linewidths of the peaks as the standard deviation of $f_{IMB,m}$. The resulting $f_{rep}$, $1093.16 \pm 0.33$ MHz, matches the value derived from the comb spectrum.

## Data availability

All the raw and processed data used in the figures in the main text and Supplementary Information are available in Zenodo repository (https://doi.org/10.5281/zenodo.6569664)[46].

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

## Acknowledgements
We would like to thank M. Asada and S. Suzuki in Tokyo Institute of Technology for valuable discussions. We appreciate Y. Takida and H. Minamide in RIKEN for their helpful discussions. This work was supported by JST ACCEL (Grant No. JPMJMI17F2 to K.T.) and JST PRESTO (Grant No. JPMJPR21B1 to T.A.).

## Author contributions
T.H. conceived the experiment, performed the measurements, and analyzed the data with input from T.A. and K.T. Y.I. performed the simulation with input from T.H. and K.T. H.I. provided the Fermi-level-managed barrier diode. T.H. wrote the first draft of the manuscript and all authors contributed to manuscript revision. All work was performed under the supervision of K.T.

## Competing interests
The authors declare no competing interests.
