## [Peer Review File · Nature Communications]

REVIEWER COMMENTS

Reviewer #2 (Remarks to the Author):

T. Hiraoka et al reported that resonant-tunneling-diode oscillator can be passively mode-locked by optical feedback and can generate a terahertz frequency comb when the round-trip cavity length is equal to an integer multiple of the RTD emission wavelength. The frequency comb was investigated using heterodyne and homodyne measurements. Modal phase relation of the comb teeth was also analyzed from the single-shot temporal waveform of the heterodyne signal. Besides, RF injection locking technique was employed to successfully stabilize the harmonic repetition frequency of the comb system. A circuit simulation model was utilized to reproduce the frequency-voltage curve, the frequency comb in the vicinity of the frequency jump, and the frequency-modulated waveform. It is an interesting paper and a lot of effort was devoted to explain the results. The manuscript can be accepted for publication after some revisions:

1. In Figure 1 (a), I would suggest the authors to label out the RTD and lens. It is easier and faster for audiences to understand the experiment setup. Another solution is to replace the figure 1 (a) using the figure S1 (in Supplementary information section 1), which better summarizes the frequency comb generation and detection. The detail of the THz lens should be added in the Methods part. In Figure 1 (g), more evidences may be required to demonstrate that this figure is the linewidth of inter-mode beating of hybrid modelocked comb system, instead of modulation signal directly from RF source.
2. The RTD is a compact device based on semiconductor technology, but in this manuscript the whole THz comb system are more than 0.5-m-long due to the external optical feedback. I don't believe this is a compact system. I would suggest the authors to remove the word "compact" in the whole manuscript.
3. In the last paragraph on page 5, the authors claimed that the mode spacing was approximately proportional to the inverse of zM , but not exactly equal to the free-spectral range of Fabry-Perot cavity due to low light return. I actually doubt this argument. Please give a detail comparison of the mode spacing and the free-spectral range by taking into account the additional optical path brought by all the optical components and frequency pulling effect?
4. In the hybrid mode-locking (RF injection locking) part on page 11, the fundamental injection locking (273.3 MHz) result was missing. It would be better to investigate the fundamental injection locking first and then compare it with the harmonic one (1.0932GHz).

5. I would suggest the authors adding the following references about mode-locking and RF injection locking of quantum cascade lasers and by C. Sirtori et al, to the introduction part and hybrid mode-locking part. In the introduction, the first and the state-of-the-art mode-locking papers of terahertz quantum cascade lasers: 1. Nature Photonics, volume 5, pages 306–313 (2011). 2. Laser Photonics Rev. 11, 1700013 (2017). In the hybrid mode-locking (RF injection locking) part on page 11, the RF injection locking and harmonic injection locking stabilization paper: 1. Laser Photonics Rev. 8, No. 3, pages 443–449 (2014). 2. Photon. Res. 9, 6, pages 1078-1083 (2021).

Reviewer #3 (Remarks to the Author):

In their manuscript entitled “Passive mode-locking and terahertz frequency comb generation in resonant-tunneling diode oscillator”, the authors describe a technique to obtain a coherent frequency comb in the THz range based on the mode-locking of a resonant tunneling diode by an optical feedback. The manuscript is well written and well referenced. Their results, a room temperature frequency comb, is certainly of great interest for various metrology and high speed communication applications and would fit well in Nature Communication. Although they clearly obtain a frequency comb, the demonstration that it originates from mode-locking is confusing and lacks clarity.

(1) The most confusing point is the protocol that the authors use to fit the heterodyne temporal waveform. The authors explained that they analyzed a 65.6 μ s long waveform to check the phase stability. They explain that the long waveform cannot be fitted because of random noise or jitter. This seems to contradict their claim of a long-term stable phase-locking. Then, they state that if the phase is random during a duration corresponding to the linewidth of the comb lines (520 ns), then there is no phase locking. So why would the waveform be cut down to sections of 164 ns, shorter than this time scale? There is no clear justification for it and I would have expected the fits to be performed at least on a 520 ns-long section. Furthermore, if the limitation of using so short sections to fit the waveform comes from phase noise and jitter, why didn't the authors performed this analysis under the hybrid mode-locking regime which stabilizes phase noise and jitter?

(2) What defines or limits the total spectral width of the comb? In Fig.1C, it seems to be roughly of the order of 10 GHz. Could this narrow width be the reason why no clear and well-defined pulses are observed in the waveforms? Is there a way to expand the comb width and how would that change the temporal content?

(3) Why did the authors choose to use an external cavity length of 500 mm? Wouldn't extending this length, thus decreasing the repetition rate, allow to separate out more clearly each "pulse" in the waveform?

(4) The authors, in addition to passive mode-locking, describe a hybrid model locking. This section in the manuscript lacks a description on this mode-locking method. I suggest the authors to briefly describe this method as it would be helpful for a broad audience.

We would like to thank the reviewers for their careful reading and insightful comments. These really help us to improve the manuscript. We have revised the manuscript to reflect their comments and suggestions. Below, we will respond to each of the comments made by the reviewers.

Summary of major changes

- Revised and newly added texts are written in red in the optional marked-up version of the manuscript. In this response letter, **the statements of changed points are written in red with the pages and lines in the marked-up manuscript.**
- Minor revision on the description of the waveform analysis to make it clearer.
- Minor revision to clarify the meaning of the harmonic frequency comb.
- Minor revision to clarify the experimental setup and results of hybrid mode-locking.
- Minor revision to organize the discussion on the effect of weak optical feedback.
- Added Supplementary Section 9-2 to show the temporal waveform corresponding to a broadband comb spectrum.

Response to reviewer #2

Remarks from reviewer #2

T. Hiraoka et al reported that resonant-tunneling-diode oscillator can be passively mode-locked by optical feedback and can generate a terahertz frequency comb when the round-trip cavity length is equal to an integer multiple of the RTD emission wavelength. The frequency comb was investigated using heterodyne and homodyne measurements. Modal phase relation of the comb teeth was also analyzed from the single-shot temporal waveform of the heterodyne signal. Besides, RF injection locking technique was employed to successfully stabilize the harmonic repetition frequency of the comb system. A circuit simulation model was utilized to reproduce the frequency-voltage curve, the frequency comb in the vicinity of the frequency jump, and the frequency-modulated waveform. It is an interesting paper and a lot of effort was devoted to explain the results. The manuscript can be accepted for publication after some revisions:

Response to Remarks from reviewer #2

We greatly appreciate the reviewer for their careful reading. We are so much pleased to know the high evaluation on our results and effort. We appreciate the reviewer for suggesting improvement and expressing his questions for our manuscript. We hope that we have responded well to all the comments and the manuscript has been improved.

Comment 2-1a

1. In Figure 1 (a), I would suggest the authors to label out the RTD and lens. It is easier and faster for audiences to understand the experiment setup. Another solution is to replace the figure 1 (a) using the figure S1 (in Supplementary information section 1), which better summarizes the frequency comb generation and detection. The detail of the THz lens should be added in the Methods part.

Response 2-1a

We thank the reviewer for suggesting the improvement of the figure. **As the reviewer's comment, we added labels on the RTD oscillator and lens in Fig. 1a and the detailed description of the THz lens in Methods (Page 17 Line 6 - 8 in the marked-up manuscript).** We would like to keep using Fig. 1a in the main text to focus on the system related to the frequency-comb generation. In addition, Fig. 1a well represents the size of the RTD oscillator, optics, and the optical system.

Comment 2-1b

In Figure 1 (g), more evidences may be required to demonstrate that this figure is the linewidth of inter-mode beating of hybrid modelocked comb system, instead of modulation signal directly from RF source.

Response 2-1b

We appreciate the reviewer for pointing out an important point to be clarified. Two pieces of evidence show that Fig. 1g is the inter-mode beating instead of the modulation signal. First, it disappeared simply by blocking the terahertz emission from the RTD oscillator. Second, as shown in Fig. S7-2c, the amplitude of this peak did not depend on the modulation amplitude below -30 dBm and even decreased with -20 dBm modulation. We added the comments on this point **in P9 L15 -19 of the Supplementary Information.**

We also realized that we might have brought up Fig. 1g too abruptly in the previous manuscript. Hence, **we added some description on the results of hybrid mode-locking in P11 L17 - P12 L1.**

Comment 2-2

2. The RTD is a compact device based on semiconductor technology, but in this manuscript the whole THz comb system are more than 0.5-m-long due to the external optical feedback. I don't believe this is a compact system. I would suggest the authors to remove the word "compact" in the whole manuscript.

Response 2-2

We appreciate this sharp comment. As the reviewer points out, although we suggested a possibility of compact feedback configuration with our simulations, our experimental work was done only with a large configuration. **Hence, we removed "compact" from the sentences regarding the advantage of the mode-locked RTD oscillator. Instead, we mention the**

compactness only when introducing the RTD oscillator or referring to the possible compact configuration shown by simulation.

Comment 2-3

3. In the last paragraph on page 5, the authors claimed that the mode spacing was approximately proportional to the inverse of z_M , but not exactly equal to the free-spectral range of Fabry-Perot cavity due to low light return. I actually doubt this argument. Please give a detail comparison of the mode spacing and the free-spectral range by taking into account the additional optical path brought by all the optical components and frequency pulling effect?

Response 2-3

We are grateful to the reviewer for expressing this natural question. However, several pieces of evidence show that the small feedback amplitude is the reason for the smaller mode spacing compared to the free-spectral range of the Fabry-Perot cavity.

At first, only considering the optical path length increased by optical components, we cannot explain the small mode spacing: Since the mode spacing Δf is 273.3 MHz, corresponding Fabry-Perot cavity length is calculated as $c/2\Delta f = 548$ mm. The measured path length between the RTD oscillator and the mirror is 500 mm, and its error would be at most ± 10 mm. Hence, we need something that increases the optical length by at least 38 mm. Among the optical components, the lens has the most significant thickness. However, its thickness is only 7 mm at the center. Considering the refractive index of the Tsurupica lens of approximately 1.5, the increase of the optical length is only 3.5 mm, which is not enough to fill the gap.

Second, the mode spacing obtained by the simulation is also smaller than the inverse of the feedback delay time used in the simulation. Hence, we can eliminate the underestimation of the optical path length from the possible cause of the small mode spacing.

The small feedback effect has been studied in the context of semiconductor lasers [D. Lenstra *et al.*, *Physica B+C*. **125**, 255–264 (1984).], and we have made a similar analytical discussion in Supplementary Section 2 (Previously numbered as Section 5). The frequency pulling effect mentioned by the reviewer might be included in this discussion.

To make these points clearer, we modified the manuscript as follows:

- We clarified the difference of the experimentally observed mode spacing and $c/2z_M$ (P5 L16 - 18.)
- We also clarified the difference observed in the simulation (P13 L7 - 10.)
- We mentioned Supplementary Section 2 and the simulation result that supports our discussion on the small feedback effect (P5 L18 - P6 L3.)
- We modified Supplementary Section 2 to discuss the mode spacing more explicitly.

Comment 2-4

4. In the hybrid mode-locking (RF injection locking) part on page 11, the fundamental injection locking (273.3 MHz) result was missing. It would be better to investigate the fundamental injection locking first and then compare it with the harmonic one (1.0932GHz).

Response 2-4

We thank reviewer for this suggestion motivated towards a better experiment. However, we believe that the modulation frequency of 1.0932 GHz is the most appropriate for our demonstration of the hybrid mode-locking for the following reasons.

First, the purpose of the demonstration is showing that we can stabilize *the repetition frequency of the frequency comb* with the bias-voltage modulation. Our frequency comb is the harmonic comb, in which the comb spacing is an integer multiple of the fundamental mode spacing. Here, the repetition frequency is defined by the separation of the harmonic-comb lines and the first inter-mode beat note frequency, but not by the fundamental mode spacing [Jaurigue *et al.*, *Phys Rev E* **93**, 022205 (2016).] We also note that there is no inter-mode beat note at 273.3 MHz. Hence, the repetition frequency is 1.0932 GHz in our frequency comb, and the modulation frequency of 1.0932 GHz is the most straightforward. To clarify these points, **we modified the introduction part of the hybrid mode-locking in P11 L12 - 16. We also clarified the concept of harmonic frequency comb, where the separation of the strongly oscillating modes, i.e., *repetition frequency*, is an integer multiple of fundamental mode spacing (P6. L3 - 5).**

Of course, investigating the effect of the various modulation signals is important to understand the hybrid mode-locking mechanisms, and the fundamental mode spacing would be one of the important modulation frequencies. However, this is a major research topic that should be presented as a separate paper because we need to investigate a vast parameter space (modulation frequencies, modulation amplitude, and possibly feedback parameters) and another discussion on the mechanism. **We mentioned this future task in P11 L10 - 12 of Supplementary Information.**

Comment 2-5

5. I would suggest the authors adding the following references about mode-locking and RF injection locking of quantum cascade lasers and by C. Sirtori et al, to the introduction part and hybrid mode-locking part.

In the introduction, the first and the state-of-the-art mode-locking papers of terahertz quantum cascade lasers: 1. Nature Photonics, volume 5, pages 306–313 (2011). 2. Laser Photonics Rev. 11, 1700013 (2017).

In the hybrid mode-locking (RF injection locking) part on page 11, the RF injection locking and harmonic injection locking stabilization paper: 1. Laser Photonics Rev. 8, No. 3, pages 443–449 (2014). 2. Photon. Res. 9, 6, pages 1078-1083 (2021).

Response 2-5

We are grateful to the reviewer for suggesting these excellent papers. These papers match the context of our manuscript well and would provide helpful information to the readers. Therefore, with a great pleasure, **we added the suggested papers in the references.**

Response to reviewer #3

Remarks from reviewer #3

In their manuscript entitled “Passive mode-locking and terahertz frequency comb generation in resonant-tunneling diode oscillator”, the authors describe a technique to obtain a coherent frequency comb in the THz range based on the mode-locking of a resonant tunneling diode by an optical feedback. The manuscript is well written and well referenced. Their results, a room temperature frequency comb, is certainly of great interest for various metrology and high speed communication applications and would fit well in Nature Communication. Although they clearly obtain a frequency comb, the demonstration that it originates from mode-locking is confusing and lacks clarity.

Response to Remarks from reviewer #3

We are so grateful for the reviewer’s cooperation and high evaluation on our manuscript and results. We also appreciate the reviewer for pointing out our unclear description. We hope we solved the confusion in the following responses.

Comment 3-1a

(1) The most confusing point is the protocol that the authors use to fit the heterodyne temporal waveform.

Response 3-1a

We greatly appreciate the reviewer for making many comments on the waveform analysis. We believe that our analysis is appropriate. However, as pointed out, the original manuscript was unclear. Hence, we refined the explanation of this analysis (P7 L4 - P9 L17). The main modification is

- Justified the dividing operation on the waveform.
- Clarified why 164 ns was chosen as the length of the short span.

Comment 3-1b

The authors explained that they analyzed a 65.6 μ s long waveform to check the phase stability. They explain that the long waveform cannot be fitted because of random noise or jitter. This seems to contradict their claim of a long-term stable phase-locking. Then, they state that if the phase is random during a duration corresponding to the linewidth of the comb lines (520 ns), then there is no phase locking.

Response 3-1b

We are grateful to the reviewer for notifying us of the confusing points of our manuscript. We considered the manuscript again to make it clearer. In our result, the initial phases φ_n vary

randomly during the long span because of the phase noise. Hence, we need to divide the long span into short spans to fit the waveform with equation (3), in which each mode is expressed with a single sinusoidal wave with a well-defined initial phase. We obtained the initial phases φ_n in each span by the fitting and calculated the relative initial phases $\Delta\varphi_n \equiv \varphi_n - \varphi_4$. In contrast to the time-varying initial phases φ_n , the relative initial phases $\Delta\varphi_n$ are found to be constant over the long span of 65.6 μs . This is evidence of the mode-locking, i.e., an effect that keeps the relation of phases constant. **To avoid this confusion, we modified the explanation in P8 L5-10 and P9 L3-8.**

Comment 3-1c

So why would the waveform be cut down to sections of 164 ns, shorter than this time scale? There is no clear justification for it and I would have expected the fits to be performed at least on a 520 ns-long section.

Response 3-1c

We greatly appreciate the reviewer for this thoughtful comment. As the reviewer writes, we must justify the length of 164 ns. Basically, we need short spans where the parameters such as frequencies and the initial phases can be regarded as constant to obtain good fitting convergence. This length is 520 ns on average (estimated from the inverse of the spectrum linewidth). However, to obtain a good convergence in as many short spans as possible, the actual fitting period must be shorter than 520 ns because it is likely to encounter phase noise in many spans of 520 ns. When we tried the fitting with the length of 520 ns, the convergence was not good in many spans. The rate of good convergence improved by shortening the fitting span, and the length of 164 ns resulted in a satisfactory convergence rate.

In addition, the length of 164 ns (more exactly, 163.84 ns) is chosen so that the number of the contained data points is equal to 2^N (here, $N = 13$), for the convenience in the analysis based on Fast Fourier Transform (FFT) algorithm. In our paper, it was beneficial in the Hilbert transform performed to the fitting result in Supplementary Section 5.

In the new manuscript, we clarified these points in P8 L5-10 of the main text and P5 L3-5 of Supplementary Information.

Comment 3-1d

Furthermore, if the limitation of using so short sections to fit the waveform comes from phase noise and jitter, why didn't the authors performed this analysis under the hybrid mode-locking regime which stabilizes phase noise and jitter?

Response 3-1d

We thank the reviewer for this natural question. We have tried the same analysis with the hybrid mode-locking case. However, the length-limitation of the fitting span is almost the same for the

hybrid mode-locked waveform and the passively mode-locked waveform. It is because the hybrid mode-locking stabilized the repetition frequency but did not stabilize the offset frequency in our experiment, resulting in the linewidths of the comb lines being almost the same as that of the passively mode-locked state (see Fig. S7-1). **To emphasize that the comb linewidth does not change by the hybrid mode-locking, we modified the description of P11 L17 - P12 L1.**

Comment 3-2

(2) What defines or limits the total spectral width of the comb? In Fig.1C, it seems to be roughly of the order of 10 GHz. Could this narrow width be the reason why no clear and well-defined pulses are observed in the waveforms? Is there a way to expand the comb width and how would that change the temporal content?

Response 3-2

We thank the reviewer for another important question regarding the spectral bandwidth. Although we could not clarify how the bandwidth is determined, we showed that it depends on the feedback condition by simulation, as shown in Fig. 4c and Fig. S10. **To clarify this point, we modified the manuscript in P15 L15 - P16 L2.**

In addition, we note that it is difficult to see a clear pulse (intensity modulation) in our mode-locked waveform. This is because the waveform is not intensity-modulated but frequency-modulated, as shown by the relative initial phases of equation (4) and a simulated waveform shown in Supplementary Section 9. **We added information of the temporal waveform corresponding to the broadband comb (Supplementary Section 9-2 and P15 L9-11 in the main text), which is still modulated only in frequency.**

Comment 3-3

(3) Why did the authors choose to use an external cavity length of 500 mm? Wouldn't extending this length, thus decreasing the repetition rate, allow to separate out more clearly each "pulse" in the waveform?

Response 3-3

We appreciate the reviewer for pointing out an insufficient description. 500 mm is the shortest cavity length in our setup that resulted in the broadest bandwidth of the comb spectrum. As the reviewer writes, the repetition rate decreased when we tried the longer cavity length. However, it also resulted in the decrease of the comb bandwidth. We believe it is because the feedback amplitude decreased due to an imperfect alignment of the THz wave. **We added this explanation in the Method on P17 L8 - 12. In addition,** it is an important future task to construct an experimental system with more efficient feedback and investigate the effect of cavity length and feedback amplitude more in detail. We mentioned this in P15 L13-15.

Comment 3-4

(4) The authors, in addition to passive mode-locking, describe a hybrid model locking. This section in the manuscript lacks a description on this mode-locking method. I suggest the authors to briefly describe this method as it would be helpful for a broad audience.

Response 3-4

We thank the reviewer for pointing out an insufficient description of the hybrid mode-locking, as reviewer #2 also pointed out. We modified the introduction part of the hybrid mode-locking in P11 L12 - 16 to describe the experimental condition.

REVIEWERS' COMMENTS

Reviewer #2 (Remarks to the Author):

The revised manuscript has been considerably improved and my concerns have been addressed. I would recommend the publication of the manuscript in nature communications.

Reviewer #3 (Remarks to the Author):

The authors have addressed my concerns. Their analysis method to demonstrate mode locking has gained clarity and their manuscript is now suitable for a broad audience. I maintain my initial assessment that this work is worth publishing in Nature Communications.

We would like to thank the reviewers for their careful reading and insightful comments through the entire the review process. These really helped us to improve the manuscript.

Summary of changes

- There is no change based on the comments from the reviewers.

Response to reviewer #2

Remarks from reviewer #2

The revised manuscript has been considerably improved and my concerns have been addressed. I would recommend the publication of the manuscript in nature communications.

Response to Remarks from reviewer #2

We greatly appreciate the reviewer for recommending the publication of our manuscript. Especially, we are grateful for their comments on several confusing points in the previous manuscript that improved the manuscript a lot.

Remarks from reviewer #3

The authors have addressed my concerns. Their analysis method to demonstrate mode locking has gained clarity and their manuscript is now suitable for a broad audience. I maintain my initial assessment that this work is worth publishing in Nature Communications.

Response to Remarks from reviewer #3

We are so grateful for their cooperation in the review process and high evaluation of our manuscript through the review process. Especially, we thank their comments on the analysis that helped us make the description much clearer.